# Reverse vaccinology-based identification of a novel surface lipoprotein that is an effective vaccine antigen against bovine infections caused by *Pasteurella multocida*

Epshita A. Islam[1ʘ], Jamie E. Fegan[2ʘ], Takele A. Tefera[3], David M. Curran[1], Regula C. Waeckerlin[4], Dixon Ng[1], Sang Kyun Ahn[2], Chun Heng Royce Lai[1,5], Quynh Huong Nguyen[1], Megha Shah[1], Liyuwork Tesfaw[3], Kassaye Adamu[3], Wubet W. Medhin[3], Abinet Legesse[3], Getaw Deresse[3], Belayneh Getachew[3], Neil Rawlyk[6], Brock Evans[6], Andrew Potter[6], Anthony B. Schryvers[4], Scott D. Gray-Owen[2]*, Trevor F. Moraes[1]*

1 Department of Biochemistry, University of Toronto, Toronto, Ontario, Canada, 2 Department of Molecular Genetics, University of Toronto, Toronto, Ontario, Canada, 3 Department of Veterinary Bacteriology, National Veterinary Institute, Bishoftu, Ethiopia, 4 Department of Microbiology, Immunology, and Infectious Diseases, University of Calgary, Calgary, Alberta, Canada, 5 Faculty of Dentistry, University of Toronto, Toronto, Ontario, Canada, 6 Vaccine and Infectious Disease Organization, University of Saskatchewan, Saskatoon, Saskatchewan, Canada

ʘ These authors contributed equally to this work.
* scott.gray.owen@utoronto.ca (SGO); trevor.moraes@utoronto.ca (TFM)

**Data Availability Statement:** All relevant data are within the manuscript and its Supporting Information files.

## Abstract

*Pasteurella multocida* can infect a multitude of wild and domesticated animals, with infections in cattle resulting in hemorrhagic septicemia (HS) or contributing to bovine respiratory disease (BRD) complex. Current cattle vaccines against *P. multocida* consist of inactivated bacteria, which only offer limited and serogroup specific protection. Here, we describe a newly identified surface lipoprotein, PmSLP, that is present in nearly all annotated *P. multocida* strains isolated from cattle. Bovine associated variants span three of the four identified phylogenetic clusters, with PmSLP-1 and PmSLP-2 being restricted to BRD associated isolates and PmSLP-3 being restricted to isolates associated with HS. Recombinantly expressed, soluble PmSLP-1 (BRD-PmSLP) and PmSLP-3 (HS-PmSLP) vaccines were both able to provide full protection in a mouse sepsis model against the matched *P. multocida* strain, however no cross-protection and minimal serum IgG cross-reactivity was identified. Full protection against both challenge strains was achieved with a bivalent vaccine containing both BRD-PmSLP and HS-PmSLP, with serum IgG from immunized mice being highly reactive to both variants. Year-long stability studies with lyophilized antigen stored under various temperatures show no appreciable difference in biophysical properties or loss of efficacy in the mouse challenge model. PmSLP-1 and PmSLP-3 vaccines were each evaluated for immunogenicity in two independent cattle trials involving animals of different age ranges and breeds. In all four trials, vaccination with PmSLP resulted in an increase in antigen specific serum IgG over baseline. In a blinded cattle challenge study with a recently isolated HS strain, the matched HS-PmSLP vaccine showed strong efficacy (75–87.5% survival compared to 0% in the control group). Together, these data suggest that cattle

**Funding:** This research was supported by equipment purchase in part by Canada Foundation for Innovation (CFI) and operating funds provided the Canadian Institutes of Health Research (CIHR) to TFM (PJT-148795) and the International Development Research Centre Livestock Innovation Fund (IDRC-LVIF - 109047). TFM holds a Tier II Canada Research Chair in the Structural Biology of Membrane Proteins and SGO holds a Tier I Canada Research Chair in Infectious Immunopathogenesis. The funders had no role in study design, data collection and analysis, decision to publish, or preparation of the manuscript.

**Competing interests:** I have read the journal's policy and the authors of this manuscript have the following competing interests: TFM, ABS, and SGO are co-authors on a patent, "Slam polynucleotides and polypeptides and uses thereof" - Patent Number: WO2017136947A1. EAI, JEF, ABS, SGO and TFM are co-authors on a provisional patent, "Veterinary vaccines and methods for the treatment of Pasteurella multocida infections in food production animals" - United States Provisional Application No. 63/332,966.

vaccines composed of PmSLP antigens can be a practical and effective solution for preventing HS and BRD related *P. multocida* infections.

## Author summary

Surface lipoproteins have been used as components of subunit-based vaccines to combat gram negative bacterial infections. A few years ago, we discovered a gene encoding a predicted lipoprotein (PmSLP) in *Pasteurella multocida* adjacent to the gene encoding the Slam translocon and showed that it was translocated to the bacterial cell surface. Since *Pasteurella multocida* has been linked to some devastating cattle diseases, hemorrhagic septicemia (HS) and bovine respiratory disease (BRD), we sought to evaluate PmSLP as a potential vaccine antigen. Upon removal of the lipid anchor, the PmSLP protein was easy to purify and could be reconstituted after lyophilization for long term storage. We investigated whether this protein could provide protection against *P. multocida* infections by establishing a mouse infection model and demonstrated that PmSLP could provide protect against invasive disease. Finally, to illustrate that the antigen could provide protection in the organism that it naturally infects, we performed large animal vaccine experiments illustrating the protective properties of PmSLP in cattle. Thus, the surface protein PmSLP provides an exciting new protein antigen that could be a cost-effective vaccine to prevent BRD and HS in cattle.

## Introduction

*Pasteurella multocida* is a gram-negative bacterium commonly found in the upper respiratory (oropharyngeal and nasopharyngeal) mucosa of a wide array of wild and domestic animals, including those relevant to the food production industry such as cattle, swine, and poultry. Disease typically arises due to invasion and dissemination of the bacterium, often under stress-inducing conditions, and can manifest as pneumonia, hemorrhagic septicemia, atrophic rhinitis, avian cholera, dermonecrosis, cellulitis, abscesses, and meningitis depending on the strain and hosts species [1,2]. *P. multocida* strains are stratified into five serogroups (A, B, D, E, and F) based on capsular polysaccharide composition and 16 serotypes based on lipopolysaccharide (LPS) antigens [3,4]. While specific virulence factors contributing to the host outcomes of *P. multocida* infection remain poorly understood, certain serogroups and serotypes have been strongly associated with specific diseases and hosts [5]. Potential virulence factors have been described including outer membrane proteins (OmpA, OmpH, OmpW), iron-regulated genes (*exbB-exbD-tonB*, *hgbA*, *fur*, *tbpA2*), and thiamine metabolism genes (*thiP*, *thiQ*) [1,6].

*P. multocida* infections have a significant economic impact on the cattle industry worldwide [7–9]. When considering cattle infections, *P. multocida* leads to two major disease outcomes, hemorrhagic septicemia (HS) or bovine respiratory disease complex (BRD), depending on the serogroup of the infecting strain. Serogroup B:2 strains have been identified as the main of cause of HS, which is a rapidly progressing, frequently fatal, septicaemic disease in cattle, bison, and buffalo. HS is prevalent in Africa and Asia and is especially devastating for small-holder farms [10]. *P. multocida* serogroup A:3 strains are known to be a key pathogen responsible for BRD [1], also known as shipping fever in feedlot cattle, a complex, multifactorial disease involving infection of the upper or lower respiratory tract and is most prominent in recently weaned calves or those that have recently arrived on a feedlot. In addition to *P. multocida*, other bacterial pathogens are implicated in BRD, including *Mannheimia haemolytica*,

*Histophilus somni*, and *Mycoplasma bovis*, while viral or lungworm infections are often concurrent with the bacterial infections. Environmental stressors including transportation, weather changes, and cold climates are considered risk factors for developing BRD. BRD is common in feedlots across North America and Europe and has been estimated to cause between 45–55% of all feedlot deaths.

*P. multocida* infections continue to be a major problem in the cattle and other livestock industries, despite the availability of vaccines for several decades. Current vaccines against *P. multocida* are either inactivated whole bacterial vaccines (bacterins) or live attenuated bacterial vaccines; however, both of these have been associated with unreliable safety profiles, poor efficacy, and relatively short duration of protection, with bacterin vaccines offering 4–6 months protection and live attenuated vaccines extending protection to approximately one year [11]. Potential explanations for the weak efficacy of existing vaccines could be due to poor immunogenicity, potential mismatch in the capsular type of the vaccine strain versus circulating strains, and the failure to induce long lasting immunity [12,13]. There has also been safety concerns related to the use of bacterins being associated with systemic reactions due to the high dose of bacterial-derived endotoxin and other products present in the doses used [13], as well as instances where live attenuated vaccines have caused chronic fowl cholera in chicken and turkey [14,15]. From the manufacturing perspective, ensuring batch-to-batch consistency and standardizing production methods is challenging, especially for autogenous bacterins, which are produced at local sites using bacterial strains isolated from infected animals, with varying growth rates, and then utilized to protect other animals from the same farm [13]. Due to the limited effectiveness and safety concerns of the existing *P. multocida* vaccines and the prophylactic and metaphylactic use of antimicrobials against bacterial pathogens including *P. multocida* in high-intensity beef production, there is ongoing concern for the development of antimicrobial resistance in pathogens, making vaccine development of critical importance [16].

The serious economic burden due to *P. multocida* infections, limited success of existing vaccines, and increasing scrutiny to antibiotic use in livestock production highlights the need for developing effective vaccines that provide long lasting protection against *P. multocida*. Various studies have proposed that conserved bacterial surface proteins, which typically play key roles in bacterial physiology and pathogenesis, could be ideal targets for the development of effective vaccines [17–20].

Bacterial surface lipoproteins (SLPs) are a class of bacterial surface proteins found on several disease causing gram-negative bacteria [20]. SLPs are characterized by a conserved N-terminal lipobox motif consisting of a cysteine residue that is post-translationally modified with an acyl lipid moiety for anchorage to the outer membrane. SLPs utilize a substrate-specific integral outer membrane protein called Surface Lipoprotein Assembly Modulator (Slam) to be translocated from the inner leaflet to the outer leaflet of the outer membrane [21]. Since SLP/Slam pairs are very often found within the same genetic cluster [22], we performed a bioinformatics search that aimed at identifying putative SLP/Slam genes in the *P. multocida* genome. This reverse vaccinology-based approach led to the discovery of a novel SLP (which we herein refer to as PmSLP) in *P. multocida*.

In this paper, we characterize PmSLP and demonstrate its utility as an effective vaccine antigen against *P. multocida* diseases in murine models of infection and protection studies in the target host, cattle.

## Results

### Phylogenetic analysis of *pmSLP* genes to investigate target conservation in BRD and HS isolates

The diverse nature of *P. multocida* and the lack of clear associations of virulence genes with specific hosts and diseases [5] prompted us to consider whether any conserved *pmSLP/slam*

genetic clusters were present, since this would suggest an essential function and provide a surface-exposed vaccine target. All available *P. multocida* genomes from the NCBI Assembly repository (287 total on April 5, 2021) were searched for the presence of the *SLP/slam* gene cluster, and the identity of the *pmSLP* gene was confirmed by the presence of an N-terminal signal peptide and lipobox motif ([LVI][ASTVI][GAS]C) [predicted using SignalP and LipoP] as well as proximity to a nearby *slam* gene.

These 287 genomes were isolated from a wide range of hosts. Bovine hosts (cows, buffalo, bison, water buffalo, yak) were the most common for these isolates (N = 116), followed by porcine (pig, boar; N = 62), avian (chicken, turkey, duck, goose; N = 41), rodent (rabbit, unspecified rodents; N = 23), human (N = 12), ovine (sheep, goat; N = 9), alpaca (N = 7), horse (N = 2), feline (N = 2), and canine (N = 1). A further 12 isolates were lacking host information. Of these genomes, the *pmSLP* gene was identified in approximately 65% of isolates (N = 189): 97% of bovine isolates (112/116), 65% of porcine (40/62), 15% of avian (6/41), 83% of rodent (19/23), 44% of ovine (4/9), 100% of horse (2/2), 50% of feline (1/2), and 42% of unknown hosts (5/12).

Phylogenetic analysis of mature PmSLP amino acid sequences indicated that variants segregate into four discrete clusters (Fig 1). Annotation of the tree to depict host, disease status, geographical location, and capsular type revealed that PmSLP in BRD and HS disease isolates were distinct. Of the 112 bovine isolates containing *pmSLP*, 76 that were serogroup A and/or were confirmed BRD isolates; 65 (85.5%) harboured PmSLP-1 (henceforth also referred to as BRD-PmSLP), 10 (13.2%) harboured PmSLP-2, and 1 harboured PmSLP-4. Of the 112 bovine isolates containing *pmSLP*, 31 were serogroup B and/or confirmed HS isolates; 30 (96.8%) harboured PmSLP-3 (henceforth also referred to as HS-PmSLP), while 1 (3.2%) harboured PmSLP-4. Of the remaining 5 *pmSLP*-containing bovine isolates, 2 harboured BRD-PmSLP, 2 harboured HS-PmSLP, and 1 harboured PmSLP-4, however these isolates lacked annotated capsule and disease type information. Finally, of the 4 bovine-associated *P. multocida* genomes that do not contain a *pmSLP*, 1 appears to be truly lacking the gene, 1 appears to have two frameshifts within the *pmSLP* locus, and the remaining 2 were of poor quality and therefore were inconclusive.

Among the 86 sequences from cluster 1, 14 were lacking the terminal 82 residues; we refer to the longer form as PmSLP-1.1 and the shorter form as PmSLP-1.2, which were identical upstream of the truncation. Within each cluster, we observed a remarkable level of sequence similarity. To compare the similarity between clusters we chose a representative sequence from each and considered only the mature polypeptide sequence. The PmSLP-1.1 representative was on average 99.99% identical to the rest of the sequences in that cluster, PmSLP-1.2 was on average 100% identical, PmSLP-2 averaged 99.99% identical, PmSLP-3 averaged 99.36% identical, PmSLP-4.1 averaged 99.79% identical, and PmSLP-4.2 averaged 97.17% identical. However, the average sequence identity between PmSLP-1 and PmSLP-3 was only 37.9%, suggesting substantial divergence of this gene among strains that cause BRD versus HS.

Interestingly, PmSLP was present in 112 of the 116 (96.6%) bovine isolates, but only found in a subset (6/41 or 14.6%) of avian isolates (found in clusters 2, 3, or 4), and a subset (40/62 or 64.5%) of in pig isolates that spanned all four clusters. This suggested that PmSLP may have a critical role to play in bovine infections, which necessitated its retention in the genome of nearly all bovine strains (for full information, see S1 Table).

Taken together, the ubiquitous presence of PmSLP in bovine isolates, along with the high sequence conservation among all BRD or all HS strains, prompted us to pursue PmSLP as a candidate vaccine antigen for the prevention of these infections.

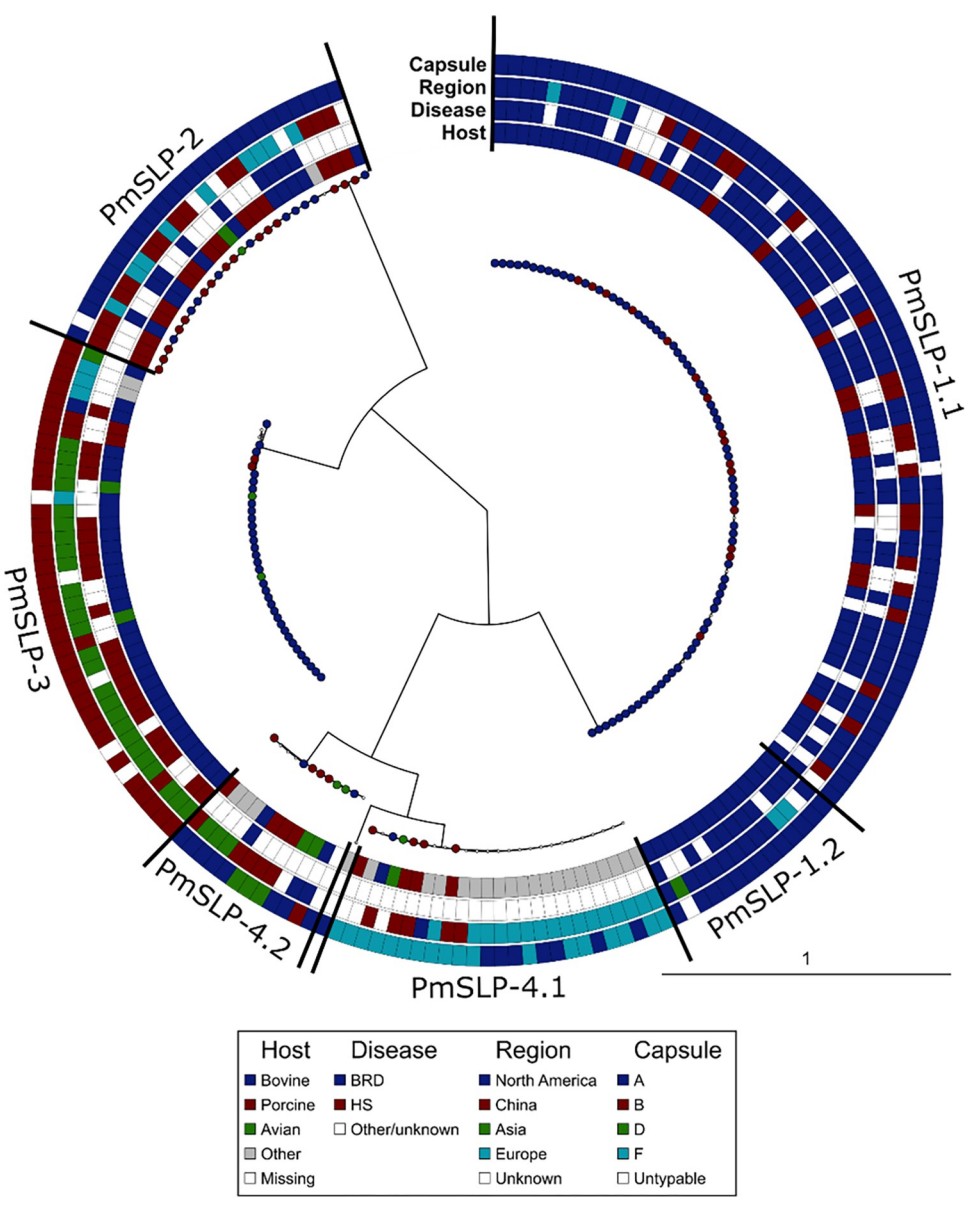

**Fig 1. Phylogenetic diversity of PmSLP mature protein sequences from published isolates of *P. multocida* including the associate capsule type, geographic location, associated disease, and animal host.** PmSLP sequences separate into four distinct clusters, two of which are further separated into two sub-clusters each.

## PmSLP-1 (BRD-PmSLP) and PmSLP-3 (HS-PmSLP) antigens protect against invasive infection by matched strains in mouse sepsis models

To develop a murine model of *P. multocida* infection, a pilot study was conducted to determine the infectious dose in naïve mice. This study showed that intraperitoneal inoculation with 100 CFU or higher of the BRD-PmSLP matched isolate (H246) was sufficient for all animals to reach clinical endpoint, while doses containing 10 CFU and 50 CFU were 80% lethal within 48h. 100% lethality was also obtained for the HS-PmSLP matched isolate (H229) with the minimum tested dose of 1000 CFU within 36h (Fig 2A and 2B). As clinical progression, symptoms, and mortality were highly consistent between the two strains, H229 was not

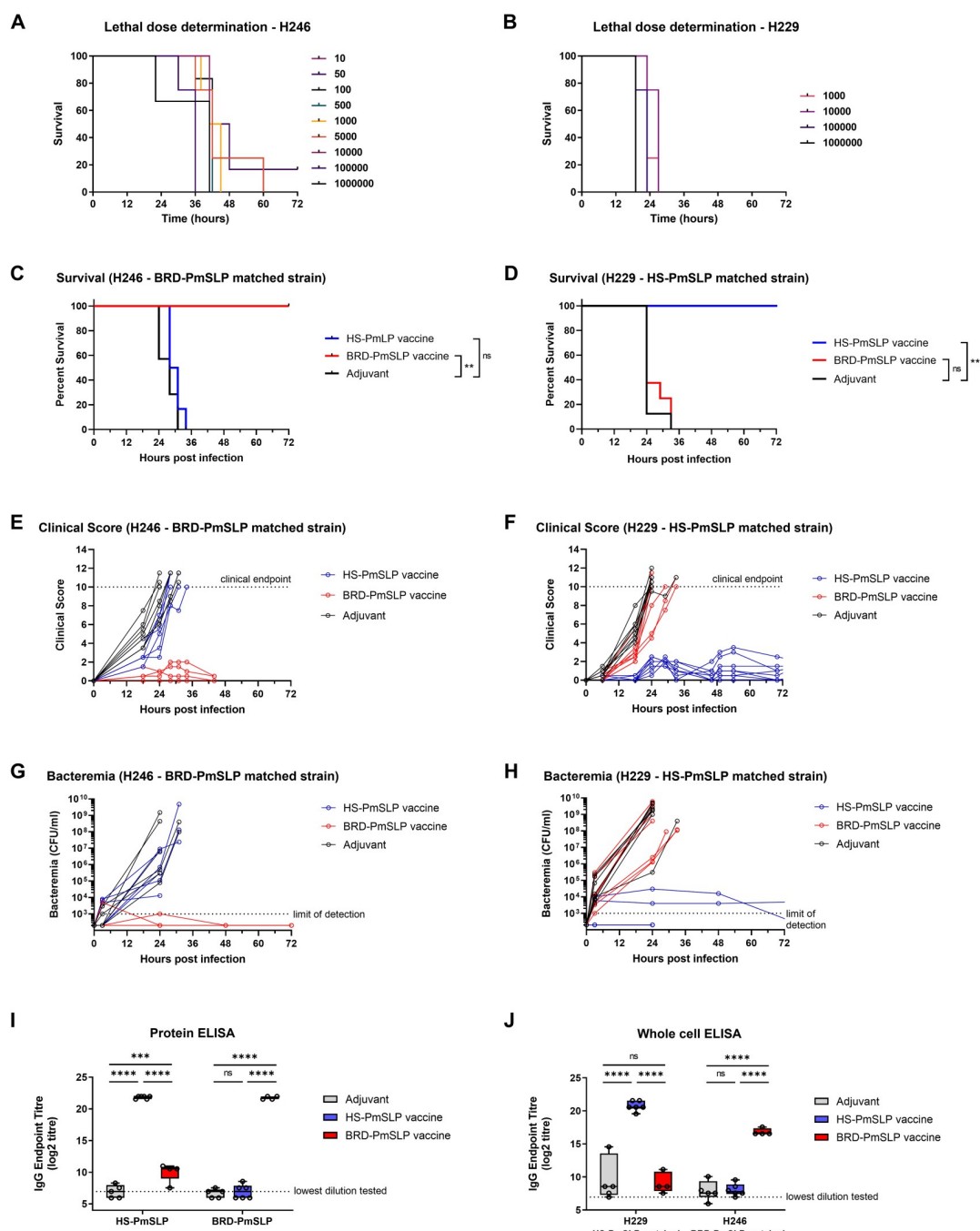

**Fig 2. Evaluation of PmSLP vaccines using a mouse sepsis model.** A. Determination of the lethal dose for a PmSLP group 1 (H246) and B. a group 3 (H229) isolate. N = 3–6 mice per group (3 mice per group for infectious dose of $10^6$ CFU; 4 mice per group use for infectious doses between 500 CFU to $10^5$ CFU; 6 mice per group for doses lower than 500 CFU). C. Survival of vaccinated and control animals after challenge with the BRD-PmSLP antigen-matched strain H246. N = 4 BRD-PmSLP, N = 6 HS-PmSLP, N = 7 Adjuvant). D. Survival of vaccinated and control animals after challenge with the HS-PmSLP antigen-matched strain H229. N = 8 per group. Log-rank (Mantel Cox) tests performed for survival curve comparisons; ****, p<0.0001; **, p<0.01; ns, not significant. E,F. Cumulative clinical scores depicted for individual animals over the course of 72 hours. Dotted lines at 10 indicate the clinical cut-off used for determining humane endpoint. G,H. Bacterial burden in tail bleeds depicted for individual animals over the duration of the challenge trial. I. Serum IgG titre in vaccinated or adjuvant control animals measured using purified BRD-PmSLP or HS-PmSLP proteins as capture. J. Serum IgG titre in vaccinated or adjuvant control animals measured using heat inactivated whole bacteria as capture. For I and J, boxplot with individual animals is shown. 2-way ANOVA with Tukey's multiple comparisons test performed on log2 transformed data to calculate p-value. ***, p<0.005; ****, p<0.0001; ns, not significant.

evaluated at doses below 1000 CFU. Infected mice developed a variety of clinical symptoms, including weight loss, dehydration, diarrhea, breathing difficulties, hunched posture and lack of movement, and notable scoring on the mouse grimace scale. Each clinical symptom was scored on a scale of 0 to 2, with a cumulative score of 10 or >20% weight loss being considered clinical endpoint, at which point mice were humanely euthanized.

Due to PmSLP-1 and PmSLP-3 only sharing approximately 37.9% sequence identity, we produced a representative recombinant PmSLP-1.1 (BRD-PmSLP) and a representative PmSLP-3 variant (HS-PmSLP) as soluble antigens in *E. coli*. Cohorts of male C57BL/6 mice were vaccinated twice 21 days apart with BRD-PmSLP or HS-PmSLP vaccines or the adjuvant only control. The efficacy against matched challenge strains was assessed using the murine infection model established in Fig 2A and 2B. Vaccines were delivered subcutaneously and consisted of 20 μg protein antigen formulated with 20% (v/v) of Montanide Gel 02 and 3 μg Poly(I:C) adjuvants in a total volume of 100 μl in phosphate buffered saline (pH 7.4). For the challenge, a Serogroup A BRD isolate, H246, was used as the PmSLP-1.1 (BRD-PmSLP) matched strain. Due to biosafety containment restrictions that precluded the use of Serogroup B HS isolates, a Serogroup A porcine isolate containing PmSLP-3, H229, was utilized as the HS-PmSLP matched strain. Infection strains are summarized in Table 1. Reflecting the conserved sequence identity among PmSLP variants, the H229 PmSLP differs from the vaccine antigen by a single amino acid residue.

For the vaccine studies, cohorts of mice were challenged via intra-peritoneal injection with $10^4$ CFU (100x lethal dose (LD)100) of each strain two weeks after the booster dose. BRD-PmSLP vaccine was 100% efficacious against the matched H246 strain (4/4 survivors) but failed to protect against the unmatched H229 strain (0/6 survivors) (Fig 2C and 2D). Conversely, the HS-PmSLP vaccine was 100% efficacious against the matched H229 strain (8/8 survivors) but provided no protection against H246 (0/8 survivors) (Fig 2C and 2D). Protected animals either had low or undetectable levels of bacteremia and showed mild or no clinical symptoms, while susceptible animals were highly bacteremic with progressive clinical symptoms and reached clinical endpoint within 32 hours post-infection (Fig 2E–2H).

Pre-challenge serum samples were evaluated using enzyme linked immunosorbent assay (ELISA) against purified PmSLP proteins or against heat killed *P. multocida* strains H246 and H229 in a whole bacterial ELISA (Fig 2I and 2J). BRD-PmSLP vaccinated animals had high serum IgG titres against purified BRD-PmSLP protein and the matched H246 whole bacteria, but these antibodies only weakly cross-reacted with HS-PmSLP protein and did not cross-react with the H229 whole bacteria. Conversely, HS-PmSLP vaccinated animals had high IgG titres against purified HS-PmSLP protein and the matched H229 whole bacteria but showed negligible cross-reactivity against BRD-PmSLP protein or the H246 whole bacteria. For each vaccine, serum was approximately 2,000- to 20,000-fold more reactive to the homologous purified protein and 150- to 6,000-fold more reactive to the matched strain versus the heterologous version. Thus, serum reactivity was predictive of the challenge outcome.

**Table 1. Challenge strains of *P. multocida* utilized in these studies.**

| Strain | PmSLP | Capsule Type | Containment Level | Challenge Study the Strain was Utilized In |
|---|---|---|---|---|
| H246 | BRD-PmSLP (PmSLP-1) | Serogroup A | 2 | Fig 2A and 2C and 2E and 2G<br>Fig 3A and 3C and 3E |
| H229 | HS-PmSLP (PmSLP-3) | Serogroup A | 2 | Fig 2B and 2D and 2F and 2H<br>Fig 3B and 3D and 3F<br>Fig 4C–4E<br>Fig 5B–5D |
| Local Ethiopian HS Strain | HS-PmSLP (PmSLP-3) | Serogroup B | 2 (in Ethiopia) | Fig 8 |
| ATCC 43017 | HS-PmSLP (PmSLP-3) | Serogroup B | 3 (in Canada) | S1 Fig |

These results suggested that recombinant PmSLP vaccines could be highly efficacious against *P. multocida* strains that harbour the *pmSLP* gene from the same phylogenetic cluster. Due to the lack of cross-reactivity and cross-protection between the two clusters, separate antigens would be required for the prevention of BRD and HS in cattle.

## Comparing the efficacy of a bivalent PmSLP-1 (BRD-PmSLP) and PmSLP-3 (HS-PmSLP) vaccine to single component vaccines

Given that BRD-PmSLP and HS-PmSLP antigens only provided cluster specific protection in the murine model, we next assessed if these two antigens can be combined within a single formulation to expand coverage. Cohorts of mice that received either BRD-PmSLP or HS-PmSLP or both antigens were subsequently challenged with the matched strains. Animals that received the bivalent vaccine were fully protected against both H246 (BRD-PmSLP matched) and H229 (HS-PmSLP matched) strains (Fig 3A and 3B). The bivalent vaccine was as effective as each single component vaccine in preventing clinical symptoms from arising (Fig 3C and 3D) and in controlling bacteremia (Fig 3E and 3F) by the matched strain. Serum IgG titres against each antigen were also quantified using protein ELISA. The bivalent vaccine was able to elicit a comparable IgG response to each antigenic component as the single component vaccines (Fig 3G and 3H).

Taken together, these results suggest that combining both antigens in a single formulation can be an effective strategy to target both BRD-PmSLP and HS-PmSLP containing *P. multocida*.

## Assessment of stability and storage of PmSLP antigens

Stability, portability, and ease of storage are important practical considerations to ensure vaccine accessibility for all end users, and is particularly vital given that *P. multocida* disease ranges from large commercial feedlots to smallholder farmers across multiple continents and varied temperature zones. While initial efficacy studies in mice showing full protection were performed using vaccines prepared with proteins that were stored at ultra-cold temperatures until formulation and then administered immediately thereafter, we investigated whether PmSLP vaccines would also be effective under practical storage and delivery settings. We focused primarily on HS-PmSLP antigen as HS is endemic in low to middle income countries and impacts smallholder farms, for whom cold chain maintenance during vaccine transport and storage may not be feasible.

We initially conducted a short-term (1–2 week) stability study to assess whether BRD-PmSLP and HS-PmSLP antigens can be lyophilized, stored at different temperatures, and still maintain structural integrity upon reconstitution. Thermal profiles of reconstituted lyophilized antigens were compared to a freshly purified reference PmSLP protein by measuring the intrinsic fluorescence given off by the tryptophan and tyrosine residues during unfolding [23,24]. The fluorescence (350nm/330nm) ratio and first derivative depicting inflection points for lyophilized antigens were comparable to the reference, suggesting that the secondary and tertiary structures were not impacted (Fig 4Ai–4Bii).

To investigate the effect of lyophilization on the protection elicited by these proteins, cohorts of C57BL/6 mice were immunized with two doses of different preparations of HS-PmSLP vaccine and then challenged with the matched H229 strain. Vaccine 1 consisted of HS-PmSLP antigen that was stored at -80˚C after purification and formulated immediately prior to the administration of each dose. Vaccine 2 consisted of HS-PmSLP antigen that was stored at -80˚C after purification, formulated prior to the first dose and stored at 4˚C for three weeks until the booster. Vaccine 3 consisted of HS-PmSLP antigen that was lyophilized and

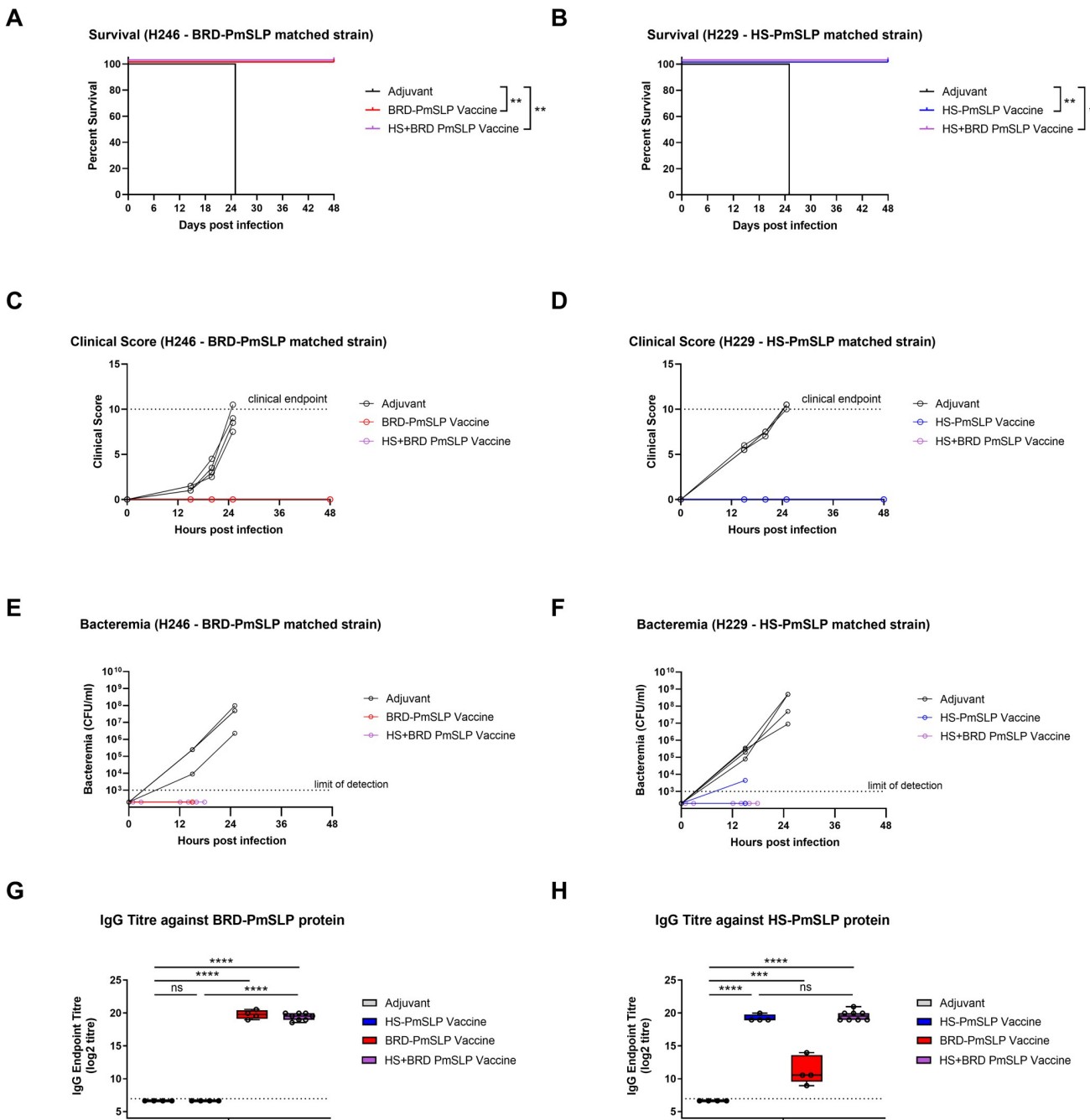

**Fig 3. Evaluation of bivalent versus single component PmSLP vaccines using a murine infection model.** Survival of vaccinated and control animals after challenge with the A. BRD-PmSLP antigen-matched strain H246 or B. HS-matched H229 strain. N = 4 per group. Log-rank (Mantel Cox) tests performed for survival curve comparisons; **, p<0.01. C, D. Cumulative clinical scores depicted for individual animals over the course of 48 hours. Dotted line at 10 indicate the clinical cut-off used for determining humane endpoint. E, F. Bacterial burden in tail bleeds depicted for individual animals over time. G, H. Serum IgG titre in vaccinated or adjuvant control animals measured using purified BRD-PmSLP or HS-PmSLP proteins as capture. Boxplot with individual animals is shown. Ordinary 1-way ANOVA with Tukey's multiple comparisons test performed on log2 transformed data to calculate p-value. ***, p<0.005; ****, p<0.0001; ns, not significant.

stored at 4˚C after purification and formulated immediately prior to each dose administration. All vaccines consisted of 20 µg protein antigen, 20% (v/v) of Montanide Gel 02, and 3 µg Poly (I:C) adjuvants, in a total volume of 100 µl in phosphate buffered saline. All three vaccinated groups were 100% protected compared to the adjuvant only group, which was highly susceptible (Fig 4C). All immunized animals showed only mild clinical symptoms (Fig 4D) and were able to control bacteremia (Fig 4E). These results indicate that lyophilized antigens retain efficacy and that the prepared vaccine can be stored for at least three weeks at 4˚C and retain efficacy.

Next, we considered whether lyophilized antigens can be subjected to long-term storage at different temperatures and still maintain structural integrity and vaccine efficacy. Aliquots of lyophilized HS-PmSLP antigen were stored at room temperature or 4˚C for 1 year and the thermal profiles compared after reconstitution (Fig 5Ai and 5Aii). Readouts for the lyophilized antigens were comparable to the reference, suggesting that lyophilization does not impact the structural properties of this antigen.

Finally, lyophilized HS-PmSLP antigen that has been stored at 4˚C for 1 year was tested for efficacy in a mouse challenge study. Vaccine 1 consisted of HS-PmSLP antigen that was stored at -80˚C after purification and formulated immediately prior to the administration of each dose. Vaccine 2 consisted of HS-PmSLP antigen that was lyophilized and stored at 4˚C for 1 year and formulated immediately prior to each dose administration. The lyophilized antigen retained full efficacy with complete protection against disease (Fig 5B and 5C), the immunized animals controlled bacteremia (Fig 5D) and the long-term stored antigen eliciting similar levels of anti-HS-PmSLP IgG as the freshly prepared antigen (Fig 5E).

Taken together, these stability studies demonstrate that PmSLP antigens can be lyophilized for ease of long-term storage and distribution and still retain full functionality. This has enabled us to overcome logistical challenges in shipping vaccines from Canada to Ethiopia for cattle studies described hereafter and is a promising characteristic for real-world vaccine deployment.

## Safety and immunogenicity of PmSLP-1 (BRD-PmSLP) and PmSLP-3 (HS-PmSLP) antigens in cattle

Given the promising efficacy of PmSLP antigens in mice and favourable stability profile, we next investigated whether these antigens are safe and immunogenic in cattle. Four separate immunogenicity studies were performed in Canada and Ethiopia, utilizing antigens under varying storage conditions, and evaluating different cattle breeds, routes of immunization, and formulation (Table 2).

Blood was sampled prior to the first dose and at regular intervals following each immunization to assess the serum antibody response. Both the BRD-PmSLP and HS-PmSLP vaccines were well tolerated by all test animals regardless of immunization route, with no serious adverse events or deaths. Transient local reactions were noted after the first dose in 50% of the animals that received Montanide Gel02 + Poly(I:C)-adjuvanted, but not Alhydrogel-adjuvanted, vaccine during Trial 4. However, 50% of animals that received Montanide Gel02 + Poly (I:C) only also developed similar local swellings after the first dose, suggesting that the reactogenicity was likely due to the adjuvant in the composition rather than the PmSLP antigen. Animals that received intramuscular immunizations (Trial 1) had no notable reactions.

Serum IgG responses were evaluated using protein ELISAs. Following immunization, an increase in BRD-PmSLP specific IgG over baseline was observed in all BRD-PmSLP vaccinated animals in both Trials 1 and 2. In Trial 1 (Canada), a significant increase over baseline reactivity was observed after 2 doses of vaccine (Fig 6Ai). In Trial 2 (Ethiopia), a small but significant

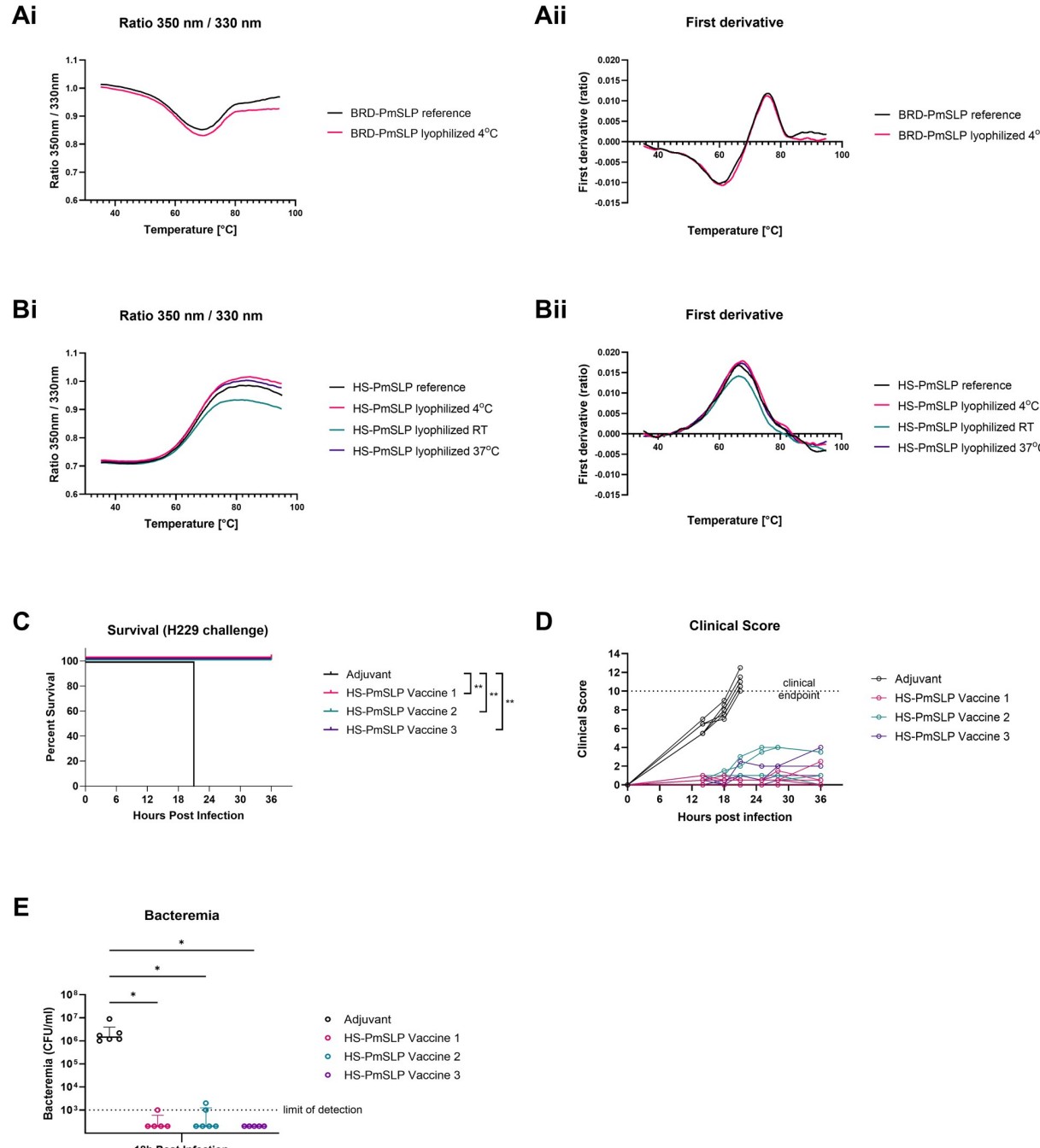

**Fig 4. Short term stability of PmSLP antigens.** A. Comparison of (i) thermal profile and (ii) first derivative of BRD-PmSLP protein lyophilized and stored at 4˚C for 7 days. B. Comparison of (i) thermal profile and (ii) first derivative of HS-PmSLP protein lyophilized and stored at 4˚C, room temperature (RT), and 37˚C for 14 days. C. Survival of mice that received different HS-PmSLP vaccine preparations after challenge with matched strain H229. Vaccine 1, HS-PmSLP antigen stored at -80˚C after purification and formulated prior to each dose administration; Vaccine 2, HS-PmSLP antigen stored at -80˚C after purification, formulated prior to first dose and stored at 4˚C until the booster; Vaccine 3, lyophilized HS-PmSLP antigen reconstituted and formulated immediately prior to each dose administration. N = 6 mice for adjuvant, N = 5 mice for each vaccinated group. Log-rank (Mantel Cox) tests performed for survival curve comparisons; **, p<0.01. D. Clinical scores of individual mice over 36 hours post intraperitoneal challenge. E. Bacteremia in tail vein sample at 18h post infection. Line at median, error bars depict interquartile range. Ordinary 1-way ANOVA with Dunnett's multiple comparison test performed to compare vaccinated groups to adjuvant control. *, p<0.05.

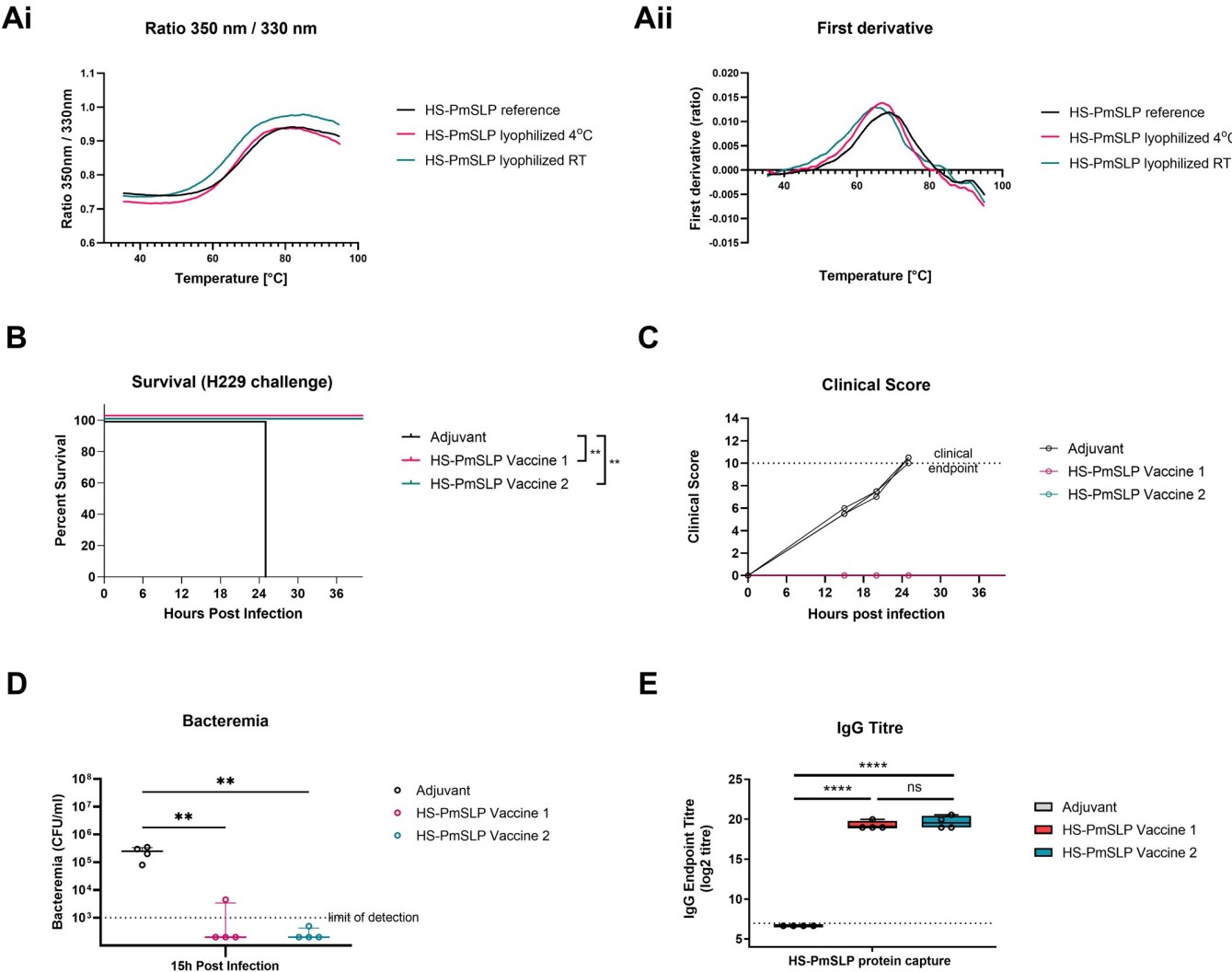

**Fig 5. Long term stability of PmSLP antigens.** A. Comparison of (i) thermal profile and (ii) first derivative of HS-PmSLP protein lyophilized and stored at 4°C or room temperature for 1 year. B. Survival of mice that received different HS-PmSLP vaccine preparations after challenge with the matched strain H229. Vaccine 1, HS-PmSLP antigen stored at -80°C for 1 year after purification. Vaccine 2, lyophilized HS-PmSLP stored for 1 year at 4°C. Both vaccines were formulated immediately prior to each dose administration. N = 4 mice for adjuvant control, N = 4 mice for each vaccinated group. Log-rank (Mantel Cox) tests performed for survival curve comparisons; **, p<0.01. C. Clinical scores of individual mice over 48 hours post intraperitoneal challenge. D. Bacteremia in tail vein sample at 15h post infection. Line at median, error bars depict interquartile range. Ordinary 1-way ANOVA with Dunnett's multiple comparison test performed to compare vaccinated groups to adjuvant control. **, p<0.01. E. Serum IgG titre in vaccinated or control animals against the purified protein antigen. Boxplot with individual animals is shown. 1-way ANOVA with Tukey's multiple comparisons test performed to compare each group to all other groups using log2 transformed data. ****, p<0.0001; ns, not significant.

increase over baseline reactivity was observed after 1 dose and was considerably enhanced after the second dose (Fig 6B). In both trials, a third dose of vaccine did not boost the response any further. Consistent with our observation in mice, there was negligible cross-reactivity against HS-PmSLP in BRD-PmSLP-immunized cattle (Fig 6Aii). Adjuvant immunized animals showed no increase in BRD-PmSLP specific antibodies over the duration of the immunization period, as expected.

An increase in HS-PmSLP specific IgG over baseline was observed for both HS-PmSLP vaccine formulations used in Trials 3 and 4. In Trial 3 (Canada), a significant increase over baseline reactivity was observed after one dose and was enhanced further after the second dose

**Table 2. Experimental layout of PmSLP cattle trials.**

| Trial | Location | Animal | Treatment Groups | Schedule/ Route/Date of Blood Collection |
|---|---|---|---|---|
| 1 | Saskatoon, Canada | 2–4 month old beef cattle (mixed breeds representative of beef cattle utilized in Canada) N = 9 per group | Vaccine: 200 μg BRD-PmSLP* + 20% (v/v) Montanide Gel02 + 30 μg Poly (I:C) in 2 mL Control: 20% (v/v) Montanide Gel 02 + 30 μg Poly (I:C) in 2 mL | 3 doses (Day 0, 21, 42) Intramuscular Bleeds (Day 0, 21, 42, 63) |
| 2 | Debre Zeyit, Ethiopia | 4–6 month old Zebu cattle N = 12–14 per group*** | Vaccine: 200 μg BRD-PmSLP** + 20% (v/v) Montanide Gel 02 + 30 μg Poly (I:C) in 2 mL Controls: 20% (v/v) Montanide Gel 02 + 30 μg Poly (I:C) in 2 mL 1% ALK(SO$_4$)$_2$.12H$_2$O in 2 mL | 3 doses, (Day 0, 21, 42) Subcutaneous Bleeds (Day 0, 21, 35, 60) |
| 3 | Calgary, Canada | 4–6 month old Holstein cattle N = 5 per group | Vaccines: 200 μg HS-PmSLP* + 20% (v/v) Montanide Gel 02 + 30 μg Poly (I:C) in 2 mL 200 μg HS-PmSLP + 0.148% Alhydrogel in 2 mL | 2 doses (Day 0, 21) Subcutaneous Bleeds (Day 0, 21, 56) |
| 4 | Debre Zeyit, Ethiopia | 1–2 year old Zebu cattle N = 8 per group | Vaccines: 200 μg HS-PmSLP** + 20% (v/v) Montanide Gel 02 + 30 μg Poly (I:C) in 2 mL 200 μg HS-PmSLP + 0.148% Alhydrogel in 2 mL Control: 20% (v/v) Montanide Gel 02 in 2 mL | 2 doses (Day 0, 21) Subcutaneous Bleeds (Day 0, 21, 35) |

*antigen shipped on dry ice and stored at -80˚C until vaccine preparation

**lyophilized antigen shipped at ambient temperature to Ethiopia and stored at 4˚C until vaccine preparation

***in the vaccinated group, all 12 animals received doses 1 and 2, but only 6 animals received a third dose of vaccine. In the control group, 10 animals received doses 1 and 2 of Montanide Gel 02 + 30 μg Poly (I:C), but 6 animals received a third dose of adjuvant only. An additional 4 animals in the control group received two doses of aluminum potassium sulfate adjuvant.

(Fig 7Ai). In Trial 4 (Ethiopia), a significant increase over baseline after one dose was seen only for the Alhydrogel formulation, and for both formulations after two doses (Fig 7B). As no increase in serum IgG titre was noted after a second booster dose in the two BRD-PmSLP cattle trials, a third dose was omitted from these studies. A significant, but very modest, increase in cross-reactive IgG was observed against BRD-PmSLP when animals were immunized with HS-PmSLP with the Alhydrogel formulation (Fig 7Aii).

Overall, BRD-PmSLP and HS-PmSLP vaccines can be safely administered to cattle via either the subcutaneous or intramuscular route and that a prime-boost regimen is able to elicit antigen specific antibodies in cattle of various ages and breeds.

## Efficacy of HS-PmSLP vaccine against invasive Serogroup B HS challenge in cattle

A blinded cattle challenge trial was conducted in Ethiopia using a locally isolated Serogroup B HS strain. The *pmSLP* gene from the challenge strain and from 8 other HS isolates in Ethiopia's National Veterinary Institute (NVI) strain collection were sequenced and were found to be identical to each other and to the vaccine antigen (See supplemental material for sequences). These *P. multocida* strains were obtained from cattle displaying clinical signs of HS from the Adea and Boset districts in central Ethiopia between November 2017 to May 2018.

Healthy Zebu breed cattle (1–2 years old) that were seronegative by hemagglutination inhibition (HI) assay against all serotypes of *P. multocida* were randomized into 4 groups of 8 animals each. 3 of these groups were immunized with blinded vaccine samples sent from Canada, and the 4[th] group received an alum-adjuvanted bacterin generated locally using the challenge strain to serve as a positive control. The blinded samples included vaccines composed of 200 μg HS-PmSLP and 20% (v/v) Montanide Gel 02 and 30 μg Poly(I:C), or 200 μg HS-PmSLP and 0.148% Alhydrogel, or only 20% (v/v) Montanide Gel 02 and 30 μg Poly(I:C). Blinded

**Ai** **Cattle trial 1 (BRD-PmSLP vaccine) - BRD-PmSLP ELISA**

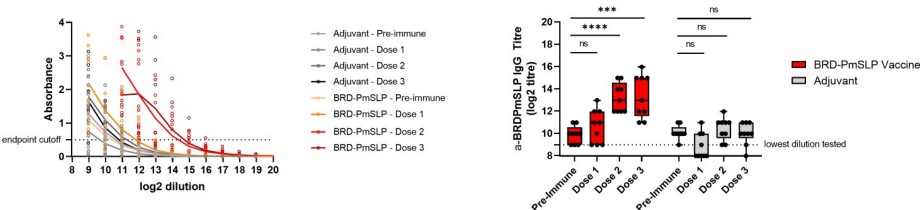

**Aii** **Cattle trial 1 (BRD-PmSLP vaccine) - HS-PmSLP ELISA**

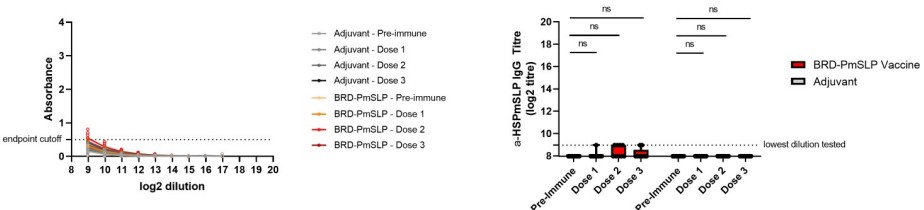

**B** **Cattle trial 2 (BRD-PmSLP vaccine) - BRD-PmSLP ELISA**

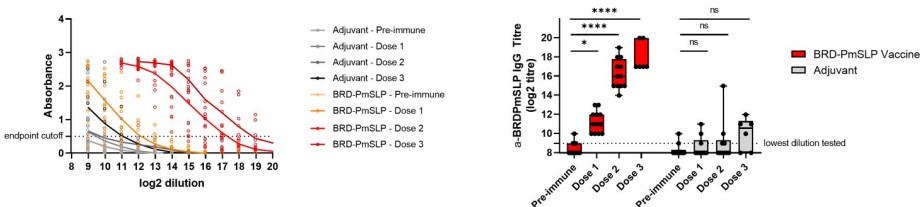

**Fig 6. Cattle immunogenicity studies using the BRD-PmSLP vaccine.** Ai, B: BRD-PmSLP antigen specific IgG detected in serum of cattle from Trials 1 and 2 respectively. Aii: Cross-reactivity of serum from Trial 1 against HS-PmSLP protein. (Left) Background subtracted absorbance readings obtained for each animal using serially diluted serum. Dotted line represents the cut-off used for calculating IgG titre. (Right) Boxplot displaying individual animals. 2-way ANOVA with Dunnett's multiple comparisons test performed on log2 transformed data to compare post vaccination titres to baseline. * $p<0.05$, *** $p<0.005$, ****$p<0.0001$, ns = not significant.

samples were sent as vials of lyophilized HS-PmSLP antigen or phosphate buffered saline (control) and the corresponding diluent containing adjuvant and were shipped at ambient temperature. Samples were stored at 4˚C after receipt, and lyophilized antigen was reconstituted in the diluent prior to each vaccine dose administration. Animals received 2 doses of vaccine or control, 21 days apart (refer to Fig 7C for anti HS-PmSLP serum IgG levels).

Approximately two weeks after the booster dose, animals were subcutaneously challenged with $4.4 \times 10^4$ CFU of the local Serogroup B *P. multocida* HS isolate, a dose that had previously been demonstrated to be lethal. In the adjuvant control group, all cattle reached endpoint within 24h. HS-PmSLP vaccinated animals had a survival rate of 87.5% (7 survivors out of 8, HS-PmSLP adjuvanted with Alhydrogel) and 75% (6 survivors out of 8, HS-PmSLP adjuvanted with Montanide Gel 02 and Poly (I:C)), while the strain-matched bacterin was fully protective (8 survivors out of 8) (Fig 8). *P. multocida* was recovered from the blood and internal organs of all the animals that reached endpoint. Surviving animals had no detectable *P. multocida* in the blood, indicating that the infection was successfully controlled (S2 Table).

**Ai**     **Cattle trial 3 (HS-PmSLP vaccine) - HS-PmSLP ELISA**

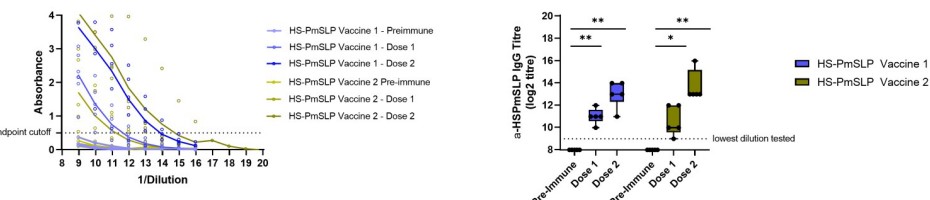

**Aii**     **Cattle trial 3 (HS-PmSLP vaccine) - BRD-PmSLP ELISA**

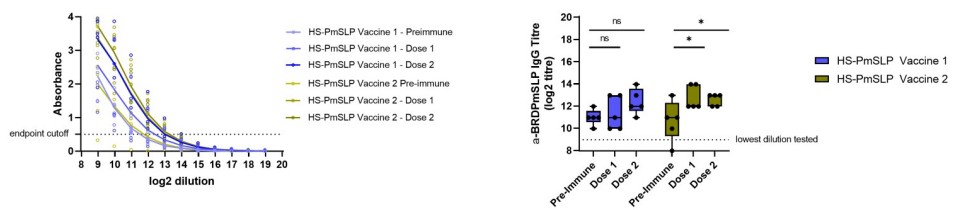

**B**     **Cattle trial 4 (HS-PmSLP vaccine) - HS-PmSLP ELISA**

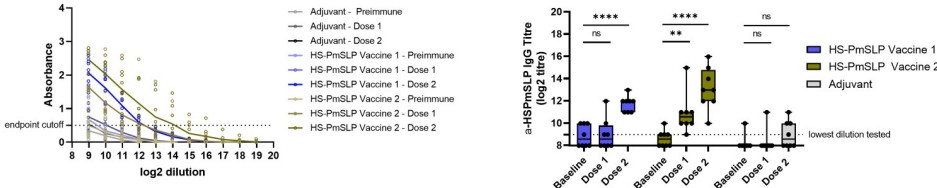

**Fig 7. Cattle immunogenicity studies using the HS-PmSLP vaccine.** Ai, B: HS-PmSLP antigen specific IgG detected in serum of cattle from Trials 3 and 4 respectively. Aii: Cross-reactivity of serum from Trial 3 against BRD-PmSLP protein. (Left) Background subtracted absorbance readings obtained for each animal using serially diluted serum. Dotted line represents the cut-off used for calculating IgG titre. (Right) Boxplot displaying individual animals. 2-way ANOVA with Dunnett's multiple comparisons test performed on log2 transformed data to compare post vaccination titres to baseline. * $p < 0.05$, ** $p < 0.01$, ****$p < 0.0001$, ns = not significant. Vaccine 1 formulated with Montanide Gel02 + Poly(I:C), Vaccine 2 formulated with Alhydrogel. Adjuvant controls received Montanide Gel02 + Poly (I:C) only.

   In summary, in this preliminary proof of concept study, two separate formulations of the HS-PmSLP antigen conferred high levels of protection against a lethal challenge by a HS-causing isolate of *P. multocida*.

## Discussion

The key to developing successful protein-based vaccines is the identification of surface-exposed antigens that are critical for bacterial survival and/or pathogenesis in the target host. Based on our analysis, PmSLP is present in a small subset of avian strains (15%) and a majority of porcine strains (65%), however it is present in 97% of bovine-associated strains evaluated to date. The ubiquitous presence of *pmSLP* in *P. multocida* strains of bovine origin, but not other host species, suggests that this surface lipoprotein may be playing a vital role during bovine

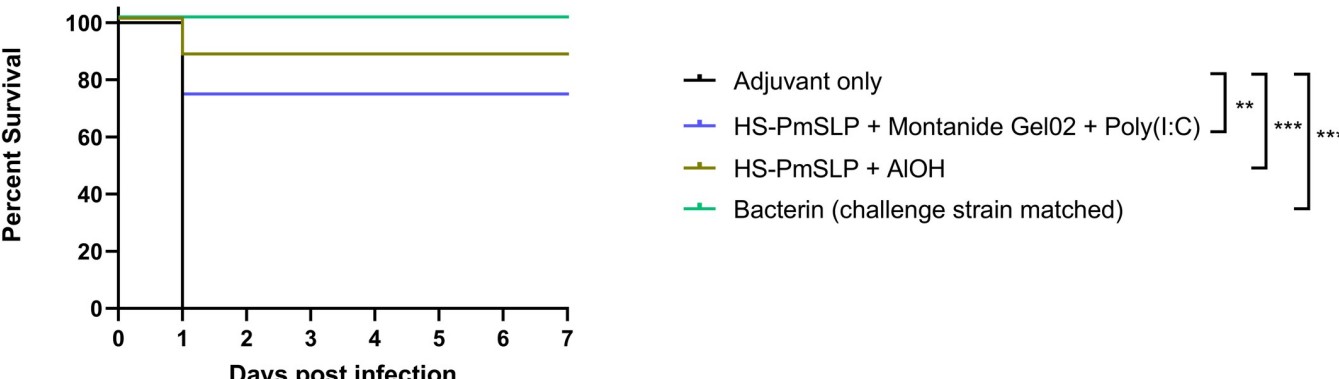

**Fig 8. Survival of Zebu cattle immunized with two different HS-PmSLP vaccine formulations or the matched bacterin following a lethal challenge with a Serogroup B HS isolate.** N = 8 animals per group. Log-rank (Mantel Cox) tests performed for survival curve comparisons; ***, p<0.005; **, p<0.01.

infections, thus necessitating its retention within the genome of bovine isolates. The existence of distinct variants of PmSLP, which segregated somewhat according to disease and serogroup, was an unexpected finding since a link between capsular type and surface lipoproteins is not typically observed [25]. Thus, future work will aim to identify the ligand(s) of BRD-PmSLP and HS-PmSLP and elucidate the functional relevance of these variants in the context of BRD versus HS infection.

This is the first study to demonstrate efficacy of a surface lipoprotein-derived antigen against *P. multocida* disease in cattle. Previously, another *P. multocida* lipoprotein, PlpE, had been shown to be efficacious against fowl cholera [26,27], and even though PlpE has been detected in bovine isolates [28], its efficacy against cattle infections has not been reported. Interestingly, PlpE is present in nearly all BRD-associated strains of *P. multocida*, but none of the HS-associated strains surveyed to date (S1 Table). While the function of PmSLP is currently under investigation, our immunization studies provide strong evidence of the utility of PmSLPs as commercially viable vaccine antigens. We have demonstrated that both BRD-PmSLP and HS-PmSLP antigens are safe and immunogenic in three separate breeds of cattle, ranging in age from as young as 2 months to as old as 2 years, and by two separate immunization routes (intra-muscular and sub-cutaneous). Notably, a 10- to 250-fold increase over baseline in anti-PmSLP IgG titres was observed with each PmSLP vaccine in cattle from both the Canadian and Ethiopian cohorts, despite potential differences in prior exposures to *P. multocida* and other pathogens endemic in these regions or differences in commensal upper respiratory tract microbiome compositions. The consistency observed in these studies provides confidence in the ability of PmSLP vaccines to elicit antigen specific antibodies in various cattle breeds at different ages.

Ultimately, the strongest evidence of the utility of PmSLP antigens came from challenge studies. Due to the cost-effectiveness and widespread use of mouse models as surrogates for vaccine evaluation [12,13], we initially used mouse models to demonstrate 100% protection against PmSLP-group matched challenge strains at infectious doses that were at least a hundred-fold higher than the LD100. Consistent with this result, 75%-87% efficacy was observed in the target species, cattle, when using HS-PmSLP vaccines in Trial 4 against a recent locally isolated HS strain in Ethiopia, under circumstances that are discussed further below. No cross-protection was observed in the reciprocal mouse studies, or in BRD-PmSLP immunized cattle during Trial 1 following challenge with an archived HS isolate from South Asia (ATCC 43017) (S1A Fig). 40% and 30% efficacy were observed after 2 and 3 doses of BRD-PmSLP vaccine in

Trial 2 following challenge with the Ethiopian HS isolate (S1B Fig). Taken together, these results suggest that low level of cross-protection may be achievable in some scenarios, but the protection is far inferior compared to the matched vaccine. Multiple confounding factors may be leading to variable degrees of cross-protection, including cattle breed, animal age, nutrition status, parasite status, previous exposure to *P. multocida*, and varying challenge isolates, however the lack/partial cross-protection, as well as the lack of IgG cross-reactivity, is not surprising considering that the BRD-PmSLP and HS-PmSLP antigens share <40% amino acid sequence identity to each other. Due to the geographical separation of capsule type and PmSLP-type present in isolates of *P. multocida*, a targeted single component PmSLP vaccine maybe the most cost-effective option in some regions. On the other hand, combining both antigens within a single formulation may be prudent in regions like China, where both Cluster 1 and Cluster 3 PmSLP-containing strains have been reported.

Another component that can be varied during formulation development is the choice of adjuvant. Our mouse and cattle studies primarily utilized either a polymer-based adjuvant called Montanide Gel 02 in combination with the TLR3 agonist Poly (I:C) or Alhydrogel. The advantage of aqueous adjuvants is that these can be included in the diluent, or may even be lyophilized with the antigen, at the final working concentration. Thus, lyophilized components and diluent can be provided in separate vials to be mixed prior to administering the dose. However, the caveat of using these aqueous adjuvants is that these may not be the most immunogenic options for cattle. For instance, a comparative study showed that a water-in-oil adjuvant emulsion adjuvant elicited higher IgG titres for a longer duration [29]. Ultimately, the choice of adjuvant would depend on the desired attributes of the end product for the target market and can be further optimised, however that was beyond the scope of this study.

In addition to cost, cold chain maintenance can be a substantial barrier to vaccine accessibility, especially for small holder farmers. Therefore, we explored the possibility of developing formulations that could be transported and stored long term at ambient temperatures. While extensive formulation work was beyond the scope of this paper, pilot biophysical stability studies were performed to demonstrate that lyophilized PmSLP antigens can be stored for at least 1–2 weeks at temperatures as high as 37˚C, and for at least one year at ambient temperature, without any significant changes in the structural integrity when reconstituted. *In vivo* studies in mice provided further support that lyophilized antigens can retain efficacy for long periods. Finally, cattle trials performed in Ethiopia utilized lyophilized protein that had been shipped at ambient temperature and spent three to four weeks in transit. Despite these challenges, the HS-PmSLP vaccines were able to provide 75%-87% protection after challenge. Although this rate was lower compared to the bacterin group, it is worth noting that these antigens had undergone a "real life" procurement scenario, as opposed to the bacterin which was prepared from the challenge strain and stored under ideal conditions. Together, these results imply that it is possible to develop PmSLP-based formulations that retain efficacy even without cold chain and in environments more akin to real world usage.

While beyond the scope of the current study, PmSLP-2 is also an attractive candidate vaccine antigen and may be necessary to elicit broad spectrum protection against BRD-causing strains of *P. multocida*. The current bioinformatic analysis of existing sequences is somewhat geographically limited, a facet which makes it difficult to predict if regions exist where BRD is caused predominantly by PmSLP-2 containing isolates. While *P. multocida* has not been demonstrated to be naturally competent, it does contain homologues of the genes known to be involved in competency in *Haemophilus influenzae* [30], therefore the potential of DNA exchange remains. Therefore, a BRD vaccine may further benefit by targeting both PmSLP-1 (referred to as BRD-PmSLP in this study) and PmSLP-2 to decrease the likelihood of vaccine escape.

While the studies presented here have focused on evaluating protection shortly after vaccination, the lack of long-term protection has been noted as an issue facing currently available vaccines. Duration of immunity studies after six-, 12-, or 24- months post vaccination will be vital in establishing the potential benefits of any PmSLP-based vaccine. Further, difficulty exists particularly with HS vaccines in ensuring two dose regimens are completed, therefore evaluating the level of protection that can be elicited via a single dose vaccine would be highly attractive.

The use of surface lipoproteins as attractive vaccine immunogens has gained traction for a number of uses in recent years, particularly as many capsular-based vaccines or attenuated vaccines offer limited breadth of protection. The use of a novel surface lipoprotein investigates here further demonstrates this class of protein as worthy of further development for inclusion in novel vaccines.

To summarize, our studies have demonstrated that PmSLP vaccines can be safe, efficacious, and practical options for the prevention of HS and BRD due to *P. multocida*. These vaccines represent a significant technological advancement over existing products in a market that has seen little to no innovation in decades.

## Methods

### Ethics statement

All mouse studies were performed in accordance with University of Toronto Animal Ethics Review Committee under protocol 20011319. Cattle trials were all performed in accordance with local regulatory bodies, including under animal use protocol 20180036 from VIDO(Vaccine and Infectious Disease Organization)-Intervac, animal use protocol AC20-0007 from the University of Calgary, and under local regulatory approval at NVI (National Veterinary Institute), Ethiopia.

### Identification of a novel surface lipoprotein

Bacterial surface lipoproteins (SLPs) are synthesized in the cytoplasm and transported to the outer membrane via a series of transport systems. The last step in this transport pathway involves an integral outer membrane protein called surface lipoprotein assembly modulator (Slam), which was originally discovered by our group in the human pathogen, *Neisseria meningitidis* (Nme). Blast searches using the *nmeSlam* gene sequence revealed the presence of *slam* homologs in several hundreds of gram-negative bacteria, including *P. multocida*. Furthermore, the presence of a conserved genome organization became apparent in several bacteria, where *slam* genes were located adjacent to putative SLP-encoding genes, as identified by an N-terminal signal peptide and lipobox motif ([LVI][ASTVI][GAS]C) [predicted using SignalP and LipoP], and a soluble protein β-barrel fold [InterPro signature: IPR or IPR011250] [22]. Analysis of the neighbouring region to the *pmSlam* gene in a commonly studied avian *P. multocida* isolate, Pm70, led to the identification of the novel SLP of unknown function, which we now refer to as PmSLP.

### Bioinformatic analysis of PmSLP sequences

Genome sequences were collected from the NCBI Assembly database, and the accompanying isolate information collected from their respective BioSample entry. tBLASTn [31] was used to identify the PmSLP sequence in each genome, and they were aligned using the E-INS-i algorithm of MAFFT v7.475 [32]. ProtTest v3.4.2 [33] was used to identify the most appropriate

evolutionary model, and the phylogenetic reconstruction was carried out by RAxML v8.2.12 [34] using the command:

raxmlHPC-PTHREADS-AVX2 -s ALIGNMENT_FILE -n PmSLP_mature -m PROTGAM-MAWAG -p 54321 -T 4

## PmSLP production and purification

Plasmids containing *pmSLP* constructs were transformed into *E. coli* T7 express via 45 second heat shock and 1 h recovery in Luria-Bertani (LB) broth (Thermo Fisher Scientific) at 37˚C shaking. Transformants were selected on LB agar with 100 μg/mL ampicillin. Multiple colonies were used to inoculate starter cultures in 20 mL LB with 100 μg/mL ampicillin and grown at 37˚C shaking for 16 h. These overnight cultures were centrifuged at 4,500 x g for 4 min and the pelleted cells were used to inoculate 2 L of LB and this larger culture was grown at 37˚C while shaking for approximately 3 h or until $OD_{600}$ = 0.5. Protein expression was induced with the addition of isopropyl β-d-1-thiogalactopyranoside (IPTG) to a final concentration of 5 mM and cells continued to grow overnight, shaking at 20˚C. Cells were pelleted at 4,500 x g and resuspended in 40 mL of lysis buffer (50 mM Tris-HCl [pH 8.0], 300 mM NaCl) with 10 mM imidazole, 1 mM phenylmethylsulfonyl fluoride (PMSF), 1 mM benzamidine, 1 mg/mL lysozyme, 0.03 mg/mL DNase I. Cells were lysed by sonication (Branson) for 2.5 minutes and centrifuged for 45 min at 30,000 x g to remove cell debris. The supernatant was passed through a 0.45 μm syringe filter and was incubated at 4˚C overnight shaking with 2 mL HisPur Ni-NTA resin (ThermoFisher Scientific). Beads were pelleted for 2 min at 700 x g, loaded onto a gravity flow column (Econo-Pac Bio Rad), and washed with 100 mL cold wash buffer (lysis buffer with 20mM imidazole). Protein was eluted in 12 mL cold elution buffer (lysis buffer with 400mM imidazole) and incubated with 2U bovine thrombin (Sigma Aldrich, cat#T4648) in dialysis against 500 mL of 25 mM Tris-HCl [pH 8.0] and 100 mM NaCl at 4˚C overnight. Dialyzed sample was incubated with 100 μL HisPur Ni-NTA resin (Thermo Fisher Scientific) and 100 μL p-aminobenamidine-agarose (Sigma Aldrich) for 1 h with shaking at 22˚C. Cleaved proteins were 0.22 μm syringe filtered, concentrated to 20 mg/mL by 10K MWCO concentrator (Thermo Fisher Scientific), and purified by size exclusion chromatography (Superdex 75 10/300 GL, GE Healthcare). For antigen studies, PmSLP protein was further purified on a strong anion exchange chromatography column (MonoQ 5/50 GL, GE Healthcare) to remove endotoxins.

## Murine infection model

In order to evaluate the potential efficacy of PmSLP-based vaccines for protection against *P. multocida*, a murine model of invasive infection was developed. For all mouse studies, mice were provided food and water *ad libitum* and were housed in individually ventilated cages in groups of 3–4 in a specific pathogen free facility and provided enrichment. All mice were handled in a biosafety cabinet. To establish the parameters for infection, 6–7 week-old male C57Bl/6 mice (Charles River) were infected with varying doses of *P. multocida* ranging from 10 CFU up to $10^6$ CFU via intra-peritoneal injection. Bacteria were grown the day of infection in RPMI at 37˚C with shaking until an $OD_{600}$ of 0.4–0.6 and diluted to the appropriate dose in 250 μl PBS (containing Calcium and Magnesium).

After intra-peritoneal infection under isoflurane anaesthesia, mice were followed for the development of clinical symptoms and were monitored for weight loss, body posture, dehydration, diarrhea, lack of movement, grooming, and for the appearance of a pinched face and were humanely euthanized once a cumulative clinical score of 10 or excess of 20% weight loss was reached. Bacteremia was monitored by tail vein bleed and plated on BHI-agar

supplemented with hemin and β-NAD (sBHI-agar) and grown overnight at 37°C with 5% $CO_2$ for enumeration. For vaccine studies, a dose of $10^4$ CFU per mouse was selected as it was far above the minimum lethal dose and provided consistent infection timelines.

For vaccine studies, C57BL/6 male mice (Charles River) received PmSLP vaccine formulations containing 20 μg of purified PmSLP protein (HS-PmSLP or BRD-PmSLP, depending on the study, or 20 μg of each HS-PmSLP and BRD-PmSLP for the bivalent study) formulated in phosphate buffered saline (PBS, Wisent) with 20% v/v Montanide Gel 02 (Seppic) and 3 μg Poly (I:C) (Invivogen) in a final volume of 100 μL. Vaccines were delivered via sub-cutaneous injection in a prime-boost schedule with the first dose given at approximately 5–6 weeks of age with the booster provided after a 3 week interval. Blood samples were taken prior to infection and were collected via saphenous bleeds. Serum was separated and stored at -20°C until analysis. Mice were challenged two weeks after the second immunization and monitored for clinical progression as above and humanely euthanized if a clinical score above 10 or >20% weight loss was reached. Challenge strains utilized in these studies are summarized in Table 1.

## Stability studies

Stability of purified PmSLP proteins stored short term (2 weeks) or long term (one year) after lyophilization at either 37°C, 4°C, or at room temperature were evaluated by thermal denaturation utilizing the Tycho (NanoTemper) [35] and compared to recently purified reference antigen. Freshly prepared PmSLP was prepared at a concentration of 100 μg/mL in buffer (phosphate buffered saline, pH 7.4). Lyophilized proteins were reconstituted at 100 μg/mL in the same buffer directly prior to analysis. This approach involves measuring the intrinsic fluorescence (detected at both 350 nm and 330 nm) given off by tryptophan and tyrosine residues as a thermal ramp is applied to the sample and the proteins begin to unfold. Changes in fluorescence signal (Ratio 350nm/330nm) indicate transitions in the folding status of a protein and the temperature at which a transition occurs is called the inflection temperature ($T_i$). Comparison of the thermal denaturation profile and Ti of samples stored under different conditions provides a rapid biophysical assessment of the structural integrity of the protein.

## Cattle studies

Four cattle experiments were performed in this study across three different locations; Trial 1 was performed at VIDO-Intervac (Saskatoon, Canada), Trial 3 was performed at the Veterinary Science Research Station, University of Calgary (Calgary, Canada), and Trials 2 and 4 were performed at the National Veterinary Institute (NVI, Debre Zeyit, Ethiopia). Cattle were sourced locally for each experiment and the breed and age utilized for each study is available in Table 2. All animals utilized in these studies were unvaccinated against HS or BRD *P. multocida* and sero-negative by hemagglutination inhibition (HI) assay for prior *P. multocida* exposure at the time of procurement. Cattle for each study were acclimated to their respective facility for a minimum of 7 days, co-mingled, and randomly separated into groups. All cattle were provided food and water *ad libitum*.

For all studies, animals were randomized into groups and immunized with either PmSLP vaccines or adjuvant only, or, for Trials 3 and 4, with a locally produced bacterin vaccine. The bacterin vaccine used was a formalin inactivated local HS strain ($2x10^9$ CFU/mL) isolated from southern part of Ethiopia containing aluminium potassium sulphate adjuvant at 1% w/v concentration of the final product. Formulations for the PmSLP vaccines used in cattle contained 200 μg of protein in 2 mL and were adjuvanted with either 20% v/v Montanide Gel 02 (Seppic) and 30 μg Poly (I:C) or 0.148% v/v aluminum hydroxide gel (Alhydrogel, Sigma-Aldrich). Vaccines were provided as either two doses (day 0, 21; Trial 3 and 4) or three doses

(day 0, 21, 42; Trial 1 and 2). For Trials 1 and 3, PmSLP proteins were stored at -80˚C and formulated prior to each dose. For Trials 2 and 4, lyophilized PmSLP proteins were sent at ambient temperature along with diluent containing PBS and the relevant adjuvant from the University of Toronto to NVI, Ethiopia and, upon receipt, stored at 4˚C until formulation and immunization. Blood samples were collected prior to each dose and, for Trial 4, prior to challenge and serum samples were stored at -20˚C until use.

For Trial 4, 14 days after the booster dose, cattle were challenged sub-cutaneously with 4.4 x $10^4$ CFU/mL of a serogroup B strain of *P. multocida* and monitored for 8 days after infection. Infection strains used for the cattle challenge studies are summarized in Table 1. Infection doses used for cattle infection studies were previously found to be lethal at the doses used here.

### Immunogenicity studies

Serum samples from either mice or cattle immunized with PmSLP-based vaccine formulations of adjuvant only controls were evaluated for the presence of anti-PmSLP antibodies via ELISA.

Mouse serum was assayed against either purified protein or inactivated whole *P. multocida*. *P. multocida* was inactivated by incubation at 56˚C for minimum of two hours and verified killed by plating on sBHI agar. Whole inactivated bacteria were added to 384-well ELISA plates (20 μL/well) at an optical density ($OD_{600}$) of 0.5 and dried overnight at room temperature. Purified protein was added to 384-well plates (20 μL/well) at a concentration of 1 μg/mL in PBS buffer and incubated overnight at 4˚C. Plates were blocked with 5% BSA and serum was added at two-fold dilutions starting at 1:500 in duplicate and incubated at 4˚C overnight. After washing, plates were probed with a 1:10,000 dilution of goat anti-mouse IgG (H&L) peroxidase antibody (Jackson ImmunoResearch Laboratories Inc.) for two hours at room temperature, washed, and then developed with tetra-methyl benzoate (TMB, Sigma-Aldrich). Reactions were quenched with 2N $H_2SO_4$ and read at 450 nm.

Cattle serum was evaluated in Canada for serum from Trials 1 and 3, or evaluated in Ethiopia with reagents, including pre-coated PmSLP plates, sent from the University of Toronto for Trials 2 and 4 due to difficulty in importing serum into Canada from foot and mouth disease endemic regions. Serum samples from cattle were tested in a consistent manner to the mouse serum samples, however plates were probed with sheep anti-bovine IgG conjugated to peroxidase (Biorad, cat # AAI23P) at a dilution of 1:10,000.

### Statistical analysis

Numerical data used for generating graphs in Figs 2–8 and S1 can be found in S3 Table. Statistical evaluations were performed in Prism 9. For all challenge studies, Log-rank (Mantel Cox) tests were performed to compare the survival curves of each individual immunized group versus the adjuvant control group. Serum IgG titres were evaluated using 1 or 2-way ANOVA, where appropriate, using log2 transformed data; Tukey's multiple comparisons test was used for comparing each group to all other groups; Dunnett's multiple comparisons test was used when comparing each group to a single control group. Bacteremia in mice was evaluated using 1-way ANOVA, with Dunnett's multiple comparison test performed to compare vaccinated groups to adjuvant control. *, $p<0.05$, **, $p<0.01$, ***, $p<0.005$; ****, $p<0.0001$; ns, not significant.

### Supporting information

**S1 Fig. Cross protection studies in cattle.** A. Survival of cattle immunized with three doses of BRD-PmSLP vaccine or adjuvant control following challenge with a South Asian Serogroup B HS isolate ATCC43017. N = 9 animals per group. B. Survival of cattle immunized with two or

three doses of BRD-PmSLP vaccine or adjuvant control following challenge with an Ethiopian Serogroup B HS strain. N = 4–6 per group. Log-rank (Mantel Cox) tests performed for survival curve comparisons; ns, not significant.
(TIF)

**S1 Table. Excel sheet containing annotations of the publicly available *P. multocida* genomes evaluated in Fig 1.**
(XLSX)

**S2 Table. Bacterial recovery and detection of challenged animals.** N denotes number of animals per group;–denotes not detected; + denotes detected; N/A denotes not applicable (no animals fell within this group); ND denotes no data.
(PDF)

**S3 Table. Excel sheet containing numerical data used to generate Figs 2–8 and S1.**
(XLSX)

**S1 Text. Protein Sequences of PmSLP from HS causing strains isolated from NVI-Ethiopia.**
(PDF)

## Acknowledgments

We would like to thank members of the Moraes, Schryvers and Gray-Owen laboratories for valuable discussions.

## Author Contributions

**Conceptualization:** Epshita A. Islam, Jamie E. Fegan, Takele A. Tefera, Regula C. Waeckerlin, Andrew Potter, Anthony B. Schryvers, Scott D. Gray-Owen, Trevor F. Moraes.

**Data curation:** Epshita A. Islam, Jamie E. Fegan, Takele A. Tefera, David M. Curran, Regula C. Waeckerlin, Dixon Ng, Sang Kyun Ahn, Chun Heng Royce Lai, Quynh Huong Nguyen, Megha Shah, Liyuwork Tesfaw, Kassaye Adamu, Wubet W. Medhin, Abinet Legesse, Getaw Deresse, Belayneh Getachew, Neil Rawlyk, Brock Evans.

**Formal analysis:** Epshita A. Islam, Jamie E. Fegan, Takele A. Tefera, David M. Curran, Regula C. Waeckerlin, Dixon Ng, Sang Kyun Ahn, Chun Heng Royce Lai, Quynh Huong Nguyen, Megha Shah.

**Supervision:** Takele A. Tefera, Andrew Potter, Anthony B. Schryvers, Scott D. Gray-Owen, Trevor F. Moraes.

**Writing – original draft:** Epshita A. Islam, Jamie E. Fegan, David M. Curran, Chun Heng Royce Lai, Scott D. Gray-Owen, Trevor F. Moraes.

**Writing – review & editing:** Epshita A. Islam, Jamie E. Fegan, David M. Curran, Chun Heng Royce Lai, Scott D. Gray-Owen, Trevor F. Moraes.

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
