## [Decision Letter · Decision Letter 0]

17 Dec 2022

Dear Dr. Moraes,

Thank you very much for submitting your manuscript "Reverse vaccinology-based identification of a novel surface lipoprotein that is an effective vaccine antigen against bovine infections caused by Pasteurella multocida" for consideration at PLOS Pathogens. As with all papers reviewed by the journal, your manuscript was reviewed by members of the editorial board and by several independent reviewers. The reviewers appreciated the attention to an important topic. Based on the reviews, we are likely to accept this manuscript for publication, providing that you modify the manuscript according to the review recommendations.

Sincerely,

David Skurnik, M.D., Ph.D.

Section Editor

PLOS Pathogens

David Skurnik

Section Editor

PLOS Pathogens

Kasturi Haldar

Editor-in-Chief

PLOS Pathogens

orcid.org/0000-0001-5065-158X

Michael Malim

Editor-in-Chief

PLOS Pathogens

orcid.org/0000-0002-7699-2064

Reviewer Comments (if any, and for reference):

Reviewer's Responses to Questions

**Part I - Summary**

Reviewer #1: Because of the absence of good commercial vaccines against Pasteurella multocida, septicemia and bronchopneumonia caused by this bacterium continues to affect cattle worldwide. This manuscript describes a new P. multocida surface lipoprotein (named PmSLP) and reports the efficacy of this lipoprotein for inducing protection against P multocida infection in mice and cattle. It also reports stability of PmSLP under different storage conditions. The studies reported were well conducted and conclusions are fully supported by the results. The manuscript is well written and easy to follow.

Reviewer #2: I would like to congratulate the authors for conducting such thorough study reflecting multi-institutional and international collaborations. The results suggest that the novel surface lipoprotein could be effectively used to reduce the burden of hemorrhagic septicemia in endemic areas. Hopefully, the authors will demonstrate a reduction of BRD burden with the use of this vaccine in feedlot cattle in the future.

**Part II – Major Issues: Key Experiments Required for Acceptance**

Reviewer #1: No major issues

Reviewer #2: (No Response)

**Part III – Minor Issues: Editorial and Data Presentation Modifications**

Reviewer #1: FYI: The absence of line numbers makes the manuscript difficult to review.

Please find below some specific comments to consider.

Introduction:

- 2nd paragraph: in the reviewer's opinion, this paragraph is substandard compared to the rest of the manuscript and should be revised. Please define each abbreviation and avoid starting a sentence with an abbreviation or a number. For example, define HS and BRD. 

- 3rd paragraph: "short duration of protection": add a reference to support your statement.

- 5th paragraph: maybe replace " increasing sensitivity" by "increasing scrutiny"

Results section:

2nd paragrah: Bovine are ruminant. Please revise.

Page 7, 3rd paragraph: "H229whole bacteria" => "H229 whole bacteria"

Figure 2A: How can you get a 20% survival with 4 mice per group? (shouldn’t t it be 25%?) Please add the number of mice per group in each figure (i.e, Fig 2, 3, 5, etc. and whenever possible) that will ease the interpretation of the results presented.

Table 2: please move to supplementary material.

Discussion: 

Page 21, 1st paragraph: "due to geographical separation of these diseases": cattle in Africa and Asia can suffer from both hemorrhagic septicemia and bronchopneumonia caused by P multocida. Thus, these diseases are not geographically separated. Please revise.

Materials and Methods:

Page 22, last paragraph: define LB.

Page 24, last paragraph: replace "four cattle studies" by "four cattle experiments" to avoid repetition. Please define HI

Page 25, 2nd paragraph: "the dose used here" => please add a reference.

Reviewer #2: 1.-Please, provide a rationale for not considering PmSLP-2 as part of the vaccines to study.

2.-Please, provide information about the clinical score used in mice.

3.-How were the animals (mice and cattle) housed and handled? Were they the experimental units? Or was there a cage/pen effect?

4.-In several plots, the axis read log2 dilution when titre is meant, change to log 2 or log 2 titre as needed. 

5.-There seems to be some mislabeling in figure 2 panels I and J and Figure 3 panel G: check the bars representing the statistical significance. 

6.-Consider support the statements in the section of stability studies (page 24) with references. 

7.-Consider substituting “highly fatal” in page 3 with “mostly fatal” 

8.-I would suggest providing more nuance to the statement that the prophylactic and metaphylactic use of antimicrobials in the cattle industry is directed against P. multocida. In reality, it is against BRD, of which P. multocida is only one of the bacterial agents. 

9.-Is there more information about the 18 isolates from ruminants? From what ruminant species?

10.-Was the proportion of animals with transient local reactions from the vaccinated groups compared with that of the control group? 

11.-For how long and at what temperature were the diluted serum samples incubated as part of the ELISA assays?

12.-In page 5, it is mentioned that PmSLP was present in 112 bovine isolates, but in a previous paragraph it is stated that 65 had PmSLP-1, 11 had PmSLP-2, and 30 had PmSLP-3. What type of PmSLP did the other 6 isolates have?

13.-The discussion should be substantially modified. The repetition of results should be kept to a minimum and no new results should be presented in this section. No figures or tables should be referenced in the discussion. The discussion of the relevant literature is limited and should be strengthened. There is no discussion of the limitations of the study. A paragraph expanding on the broader impact of this study and how it may benefit further research and development of vaccines against other bacterial agents is warranted. 

14.-I consider a section of statistical analysis should be added to the methods.

PLOS authors have the option to publish the peer review history of their article (what does this mean?). If published, this will include your full peer review and any attached files.

Reviewer #1: No

Reviewer #2: No

Figure Files:

Data Requirements:

Reproducibility:

References:

---

## [Editor Report · Decision Letter 1]

27 Feb 2023

Dear Dr. Moraes,

We are pleased to inform you that your manuscript 'Reverse vaccinology-based identification of a novel surface lipoprotein that is an effective vaccine antigen against bovine infections caused by Pasteurella multocida' has been provisionally accepted for publication in PLOS Pathogens.

Best regards,

David Skurnik

Section Editor

PLOS Pathogens

Kasturi Haldar

Editor-in-Chief

PLOS Pathogens

orcid.org/0000-0001-5065-158X

Michael Malim

Editor-in-Chief

PLOS Pathogens

orcid.org/0000-0002-7699-2064
---

## [Editor Report · Acceptance letter]

20 Mar 2023

Dear Dr. Moraes,

We are delighted to inform you that your manuscript, "Reverse vaccinology-based identification of a novel surface lipoprotein that is an effective vaccine antigen against bovine infections caused by Pasteurella multocida," has been formally accepted for publication in PLOS Pathogens.

Best regards,

Kasturi Haldar

Editor-in-Chief

PLOS Pathogens

orcid.org/0000-0001-5065-158X

Michael Malim

Editor-in-Chief

PLOS Pathogens

orcid.org/0000-0002-7699-2064